# Compete or Cooperate? Goal Orientations and Coworker Popularity in the Knowledge-Sharing Dilemma

**DOI:** 10.3390/bs14030250

**Published:** 2024-03-19

**Authors:** Heesun Chae, Inyong Shin

**Affiliations:** Division of Business Administration, Pukyong National University, Busan 48513, Republic of Korea; hschae@pknu.ac.kr

**Keywords:** goal orientation, knowledge sharing, coworker influence, coworker popularity, trait activation theory

## Abstract

Focusing on two goal orientations (the learning and performance prove-goal orientation), this study proposed a different mechanism for dealing with the knowledge-sharing dilemma. We analyzed data from 257 employee–coworker dyads, finding that the learning goal orientation positively affected knowledge sharing, while the performance prove-goal orientation negatively affected knowledge sharing. In addition, highlighting the importance of coworker influence, our analysis showed that coworker popularity served as social cue to boost the main effects of knowledge sharing. Specifically, the positive relationship between the learning goal orientation and knowledge sharing and the negative relationship between the performance prove-goal orientation and knowledge sharing were stronger when coworker popularity was higher. These findings contribute to articulating theoretical directions at the individual level for addressing the dilemma associated with knowledge sharing. Furthermore, they offer practical implications by emphasizing the ongoing importance of considering the influence of coworkers, who serve as crucial exchange partners during task execution.

## 1. Introduction

Knowledge is the source of any organizations’ competitive advantage and the driving force of its values [1]. Since knowledge resides inherently within individuals, whether an organization can utilize it effectively depends on the people who create, use, and share it [2]. Consequently, individual employee knowledge sharing, defined as “the individual behavior of sharing organizationally relevant information, ideas, suggestions, and expertise with one another”, (p. 65) [3] is the critical starting point for the success of an organization’s knowledge management [4,5]. However, individual knowledge sharing is a fragile process, and its potentially significant costs can even offset its potential benefits [6,7]. Sharing and integrating useful knowledge on the cooperative side can increase overall benefits for exchange partners and thereby help recipients save time and improve results; however, such contributions may also expose personal vulnerabilities by revealing the sources of one’s competitive success, potentially limiting one’s ability to outpace others. Facing this dilemma, individuals ultimately need to decide whether to cooperate or compete.

Research on knowledge sharing in the past has predominantly concentrated on social and organizational environments, such as leader support, incentive structures, and organizational climates, as mechanisms to encourage knowledge sharing within contexts containing these challenging elements [8]. Nevertheless, there is a dearth of research on individual-level factors that contribute to fostering knowledge-sharing behavior when addressing knowledge-sharing dilemmas [7]. Indeed, while some studies have examined variables related to individual characteristics, such as goal orientation, a proactive personality, self-efficacy, exchange ideology, evaluation apprehension, and conscientiousness, systematic research incorporating motivation-based individual variation is still lacking [5]. While research on learning goal orientation (LGO) has predominantly concentrated on its positive aspects, there is a notable scarcity of studies delving into knowledge sharing and performance goal orientation (PGO) [4]. Moreover, the existing research has unveiled inconclusive patterns, indicating a deficiency and inadequacy in this particular domain [8]. In sum, previous studies have highlighted the need for more systematic and in-depth investigations of the impact of performance goal orientation on knowledge sharing, operating in parallel with learning goal orientation (LGO). Therefore, applying cooperation and competitive frameworks, we set out to comprehensively explore the effects of two different dimensions of individuals’ motivational characteristics, goal orientations, on knowledge sharing.

These two goal orientations may play distinct roles in determining how employees evaluate the costs and benefits of sharing knowledge. The cooperative dimension of such knowledge sharing refers to prosocially producing value for others and collectively using shared knowledge to pursue common interests. Individuals with an LGO value active cooperation. Perceiving coworkers as allies [9,10], they seek to both enhance learning by sharing knowledge and expand the pool of expertise by receiving feedback and embracing future learning opportunities. As a result, they may cooperate with their colleagues and act more altruistically and benevolently [10,11]. Meanwhile, the competitive dimension refers to the application of valuable skills and knowledge to achieve superiority and use shared knowledge to make private gains in an attempt to outperform others [12]. Individuals with a performance prove-goal orientation (PPGO) are highly interested in enhancing their superiority and reputations [10,13,14,15,16]. As a results, they may engage in more competitive behaviors in relation to their coworkers [9,17,18].

Furthermore, since research has suggested that contextual factors may alter actual knowledge sharing [19] and because knowledge sharing behaviors arise within relationships between givers and recipients of shared knowledge, we suggest that situational factors—and particularly recipient characteristics—influence the effects of goal orientation on knowledge sharing. In particular, the recent trend toward flattering organizational structures and blurring hierarchical distinctions has made interactions among colleagues more frequent. Such circumstances lead to an increased emphasis on knowledge sharing as an important behavior among coworkers who hold similar positions and perform similar tasks. Trait activation theory [20] also identifies coworkers as important situational variables in the expression of individual dispositions. Surprisingly, however, explanations of knowledge sharing in the context of relationships with coworkers are exceedingly rare. Specifically, studies have investigated the moderating effects of coworker self-efficacy on the relationship between members’ self-efficacy and knowledge sharing [21], the moderating effects of the tacit knowledge of knowledge recipients on the relationship between coworker support and knowledge sharing [22], and the influences of the characteristics of knowledge recipients and knowledge sharers (such as learning attitudes and personal relationships) on knowledge sharing [23]. In this study, we regarded coworkers as situational cues that accentuate or constrain the impacts of different goal orientations on knowledge-sharing behavior by making knowledge providers more cooperative or competitive, thereby affecting whether they share knowledge with coworkers. Specifically, we investigated interactions in which higher coworker popularity triggers the cooperative characteristics of LGO and the competitive characteristics of PPGO.

In summary, we hypothesized that the two facets of goal orientation have different effects on employees’ knowledge sharing behaviors and that these effects are influenced by the social context. Specifically, we focused on the cooperative and competitive mechanisms for dealing with dilemmas in these two goal orientations. In addition, emphasizing the importance of coworker influence, we examined trait-relevant situational cues to identify which specific coworker characteristics cause individuals’ attributes to activate individual motivational traits. We employed field data obtained from 257 employees and their coworkers to validate our theoretical framework.

## 2. Literature Review and Hypothesis Development

### 2.1. Goal Orientation and Knowledge Sharing

An interesting characteristic of knowledge is the fact that its value grows when it is shared. The costs of initiating knowledge exchange can include a loss of resources, the effort of performing a behavior, or simply opportunity costs. Ultimately, individuals have different perceptions of sharing knowledge, and these different perceptions may significantly impact their strategies for dealing with the conflict between cooperation and competitiveness.

Whether an individual decides to compete or cooperate comes down to reasoning and calculation. Individuals may perceive the costs of sharing ideas with a community as higher or lower depending on personal characteristics that reflect their individual propensities to cooperate or compete. Goal orientations are motivational orientations that regulate where individuals direct attention when approaching, interpreting, and responding to achievement situations [24].

While LGO involves advancing abilities by mastering new skills and circumstances, PGO involves proving and showing off one’s abilities [25,26]. Scholars have identified two dimensions of PGO [27]. While individuals with PPGO concentrate on attaining success, individuals with performance-avoidance goal orientation (PAGO) concentrate on avoiding failure [27]. Individuals with PPGO are not necessarily dysfunctional, but individuals with PAGO are dysfunctional. PAGO has been consistently linked to maladaptive forms of engagement and processing, as well as lower achievement; this means that it is associated with negative outcomes, such as anxiety, distraction, withdrawal, maladaptive performance strategies, and feelings of helplessness [28]. Thus, for PAGO, no inconsistent pattern requires further explanation. Moreover, our primary objective was to compare and evaluate the performance of LGO and PPGO in solving optimization problems associated with competition and cooperation. Thus, given the scope of and available resources for this research, as well as our aim of conducting a focused and in-depth analysis, we decided to concentrate on these two methods.

### 2.2. The Cooperative Side of Goal Orientation: LGO and Knowledge Sharing

Under conditions of cooperation, defined as “prosocial behavior performed for the common benefit of the donor and the recipient”, individuals’ behaviors are characterized by trust, commitment, reciprocity, and the use of coordination to achieve results [29]. In cooperative situations, employees help each other because they believe that their goals are positively connected [30]. Individuals with LGO take this approach—adopting a more benevolent manner in social exchange situations [10,31]. Research has revealed a high correlation between LGO and organizational citizenship behaviors, such as providing help [32] and engaging in volunteer activities [33], which are driven by prosocial values and altruism. Duda and Nicholls’ [34] notion that learning goals are associated with the belief that success and development are achieved through collaboration with others, learning from mistakes and difficulties, and exertion of effort may suggest that they would be positively linked to the tendency to trust one’s colleagues as partners who can help one understand and overcome problems and weaknesses. Individuals with LGO may perceive coworkers as allies in their efforts to enhance their own learning by expanding the pool of knowledge, which highlights the benefits of actively cooperating with others [35].

Individuals with LGO tend to seek cooperative interactions when they believe such interactions might contribute to their learning [36]. They view knowledge sharing as an opportunity to both enhance their knowledge and establish new knowledge sources for the future. Cooperation implies that one party gives up some immediate benefits in the hopes of receiving a later payoff. Individuals only engage in altruistic acts of cooperation where they calculate that said acts will improve their chances of survival. The traditional presumption has been that altruism is necessarily related to direct reciprocity, implying that following repeated acts of altruism, a donator comes to value the benefit to themselves of reciprocated acts of altruism [37,38].

Cooperation is motivated by a desire to maximize both one’s own and the others’ outcomes [39]. Individuals with LGO actively share knowledge because they perceive knowledge sharing as a learning process that develops their capabilities by giving them a chance to articulate and transfer the knowledge they possess [40]. Knowledge should be codified or transferred before it is shared with others, since sharing knowledge involves exerting the effort necessary to separate resources from their sources [41,42]. For example, when employees attempt to share knowledge but are unsure that they can comprehensibly transfer it to their coworkers, they are more likely to use knowledge sharing as an opportunity to deepen their own understanding and find better ways to explain it before they share it [43].

Moreover, individuals with LGO anticipate that sharing knowledge with colleagues will prove worthwhile because it will serve as another learning opportunity [44] and might thereby eventually help them to improve their skills. Social exchange theory [45] views sharing knowledge as an investment aimed at establishing good learning opportunities in return. For instance, in the knowledge sharing process, individuals can receive feedback about their knowledge [26] and engage in debates about work-related issues [46], which can serve as another source of learning. Correspondingly, Poortvliet and his colleagues [35] showed that learning goals foster a strong reciprocity orientation; sharing valuable information results in information exchange between partners. These exchange orientations drive individuals to provide exchange partners with better knowledge. Therefore, our first hypothesis was as follows:

**Hypothesis 1.** 
*LGO is positively related to knowledge sharing.*


### 2.3. The Competitive Side of Goal Orientation: PPGO and Knowledge Sharing

Competition is motivated by the desire to maximize one’s own outcomes relative to those of others [29]. Under conditions of competition, defined as “the desire to win in interpersonal situations”, calculation, bargaining, maneuvering, and the use of power to achieve results are characteristic behaviors [47]. Individuals with competitive orientations are interested in their own contributions and finding ways to differentiate their performance from others [48]. A meta-analytic review revealed a positive association between PPGO and competition [17,49], suggesting that PPGO may lead participants to perceive others as threats [50]. Such a competitive orientation instigates social comparison processes [18].

Individuals with PPGO tend to concentrate on comparing themselves to others to highlight their competence [25,49]. Their resulting need to manage how others perceive them while performing their jobs generates elevated self-threat levels [13,51]. Individuals with PPGO think that they deserve good treatment but view whatever others receive negatively. Their interest in performing well is motivated by extrinsic concerns, such as winning competitions, rather than intrinsic involvement in the work itself. Accordingly, individuals with PPGO tend to regard coworkers as competitors or competitive exchange partners. In turn, they see little reward in sharing their knowledge or believe such efforts involve the wasteful reinvention of their uniqueness or superiority.

People with PPGO strategically deceive exchange partners to ensure that they maintain a performance advantage [18,31]. Dietz and his colleagues [13] showed that individuals with PPGO had stronger intentions to deceive in achievement settings. They are afraid of losing their unique value by sharing knowledge and potentially damaging their reputations if coworkers judge their shared knowledge as unsound or irrelevant [3,16,52]. This fear can discourage knowledge sharing. In addition, Poortvliet his colleagues [35] found that individuals with PPGO have a stronger exploitation orientation. To prevent others from benefiting from their knowledge, they try to share as little useful knowledge as possible. They tend to keep their knowledge to themselves because sharing know-how with others hinders their efforts to achieve superiority. Their belief that successful knowledge transfer boosts competition and creates stronger competitors prevents them from devoting attentional resources to sharing their know-how.

According to resource allocation theory, individuals have fixed attentional resources to allocate to various job-related elements [53]. Given these limited attentional resources, PPGO employees generally devote their energy to in-role job factors, since they tend to regard satisfying a given role requirement as a criterion for competitiveness [24,54]. They apply themselves to surpassing others and receiving favorable evaluations from their organizations’ reward systems [19]. Consequently, they tend to view knowledge sharing as a practice that diminishes the time and effort they can devote to performing in-role jobs. Instead, they prefer to make personal gains and focus on achieving results that exceed given performance goal expectations [43,54]. Thus, individuals with high PPGO place a low value on knowledge sharing and, in turn, share less knowledge with others. Therefore, our second hypothesis was as follows:

**Hypothesis 2.** 
*PPGO is negatively related to knowledge sharing.*


### 2.4. Coworker Influence as a Situational Cue: Coworker Popularity

Drawing upon an interactionist perspective [55], which posits that the impacts of individual characteristics can vary based on the context, this study investigated situational factors influencing the effects of goal orientation on knowledge sharing. Employing an interactionist framework, such as trait activation theory [56], the research explored how situational cues, relevant to specific traits, influence the expression of individual traits, as theorized by trait activation theory [56]. In the knowledge-sharing context, a coworker, a recipient of knowledge, can be a critical situational factor that influences the effects of individual goal orientation on knowledge-sharing behaviors. Coworkers are social-comparison targets and important sources for obtaining valuable knowledge [57]. Social comparison increases in dyadic relationships where individuals are more similar [58]. Employees compare themselves to their coworkers in several respects, including popularity and ability.

We focused on the coworker characteristic of coworker popularity. Popularity is a form of social stratification, more highly related to social impact than social preference [59,60]. Popular coworkers are at the center of organizations’ communication networks and have frequent interactions with many people in the workplace [61,62]. An awareness of coworkers’ different social influences in the workplace could have implications for employees’ willingness to cooperate with coworkers [58].

### 2.5. Moderating Effects of Coworker Popularity

Based on trait activation theory [56], we expected the socially integrated position of high popularity coworkers to trigger the cooperative characteristics of LGO in social dilemma contexts. LGO has a positive relationship with prosocial goals [10,63], generating positive attitudes toward popular coworkers, who have high levels of access to valued resources. People with LGO have positive attitudes toward cooperation when they perceive it as an opportunity to enhance their own learning. They view helping others as an opportunity to improve by increasing their learning and understanding [32].

Robustly connected with other team members, coworkers positioned at the core of a network would perceive themselves as providers of increased learning opportunities. Due to their extensive interactions, coworkers in central network positions tend to be linked with many people in their teams, giving them access to valuable resources, such as knowledge [61,62]. Thus, the energy and time required for individuals with LGO to utilize popular coworkers’ connections to various experts in organization networks to increase their learning opportunities are minimal, which makes cooperation more beneficial for them. Additionally, members characterized by LGO exhibit high openness to new experiences [19,27] and persistence when facing difficulties [64,65]. Individuals with learning goals adopt a solutions-oriented approach toward problem solving. They also perceive errors as a natural part of the learning process [25,66,67]. When learning-oriented people encounter obstacles while explaining their know-how to coworkers in the process of solving difficult technical or social problems, they tend to deal with these challenges by investing additional effort [68,69]. This means that they are willing to devote additional effort and time to helping popular coworkers by sharing their knowledge. Coworkers located at the network’s center are suitable collaborators and are optimally positioned as primary sources of new learning opportunities. Interactions with them engender fresh information and learning prospects, motivating LGO individuals to invest more resources, even when encountering challenging tasks or learning opportunities. Therefore, individuals possessing these characteristics are likely to be perceived as more appealing collaborators, meaning that coworker popularity is a situational factor that can enhance knowledge-sharing behaviors among employees with LGO. Therefore, our third hypothesis was as follows:

**Hypothesis 3.** 
*Coworker popularity moderates the relationship between LGO and knowledge sharing, such that the positive relationship is stronger with high coworker popularity.*


Consistent with trait activation theory [56], we expected that the negative relationship between PPGO and knowledge sharing would be strengthened when coworkers are popular. Specifically, relative status comparisons among colleagues in the workplace trigger a competitive mindset for PPGO individuals, leading them to overweight the costs of sharing their knowledge. When coworkers are highly popular, feelings of competition may increase, making PPGO individuals less willing to engage in task-related cooperation with them. Employees who receive acceptance and admiration from their peers tend to be perceived as more influential, because their peers are more receptive to their influence attempts [70]. Therefore, interacting with highly popular coworkers could threaten the self-worth of PPGO individuals. High-PPGO individuals are deeply interested in how others perceive them and may be more likely to protect and seek to enhance their self-worth [13,18,36]. Popular individuals within workgroups are characterized as being “socially preferred and visible” [71] (p. 163) and are typically “widely accepted” by their coworkers [61] (p. 21). Coworkers with higher popularity levels often engage in increased interpersonal interactions, leading individuals who become aware of their reputations to actively seek their assistance [72]. Thus, popular coworkers increase PPGO individuals’ fear of losing their superiority or reputations because PPGO individuals perceive incomplete achievements or losing status as reflecting a lack of ability and failure [14,52].

Coworkers in popular positions, benefiting from access to a wealth of diverse information sources, are likely to possess substantial knowledge already. Thus, the knowledge PPGO individuals attempt to share often duplicates existing knowledge. This situation can undermine the abilities of PPGO individuals to highlight their unique capabilities or the superiority of their existing skills in other contexts. Consequently, sharing knowledge with popular coworkers is risky for PPGO individuals. Therefore, our fourth and final hypothesis was as follows:

**Hypothesis 4.** 
*Coworker popularity moderates the relationship between PPGO and knowledge sharing, such that the negative relationship is stronger with high coworker popularity.*


## 3. Methods

### 3.1. Sample and Data Collection

We designed the survey by separating the response sources for the outcome and predictor variables, thus avoiding common method bias [73]. Then, we distributed survey packets, which included an employee survey and a coworker survey, to the focal employees. Focal employees completed their survey, while focal employees’ coworkers performing similar tasks and work processes completed the coworker survey. Once the focal employees and coworkers had finished the surveys, they were asked to put them in a provided envelope, seal them, and return them to the researcher.

To enhance the statistical robustness and generalizability of the study findings, it is imperative to collect data from employees representing diverse industries, departments, and functions. We conducted the survey through a variety of South Korean companies spanning electronics, telecommunications, manufacturing, and construction design. The survey participants were selected from various departments and functions, including sales, human resource management, finance, research and development, production, and quality control. The survey received endorsement from top management within these companies and was conducted for one month, spanning September 2023.

Of the 280 full-time employees who received surveys, 275 submitted self-reported responses. In a separate questionnaire, we obtained coworker ratings from 270 of the 280 respondents. Overall, we gathered 257 pairs of completed matched responses. Academic conventions suggest a sample size 5–10 times the total number of items used in the survey [74]. The measurement questionnaire employed in this study, including control variables, consists of a total of 28 items. In consideration of this, the sample size for this study was 257 pairs, exceeding the recommended threshold (140–280) for statistically validating the research model.

Of the final sample of 257 focal respondents, the average age of the employees was 36.7 years (SD = 8.6) and the average tenure with the coworker was 3.3 years (SD = 4.1). The sample included 147 men (57.2%) and 110 women (42.8%). The level of education varied from high school graduation to a Ph.D. degree; 4.3% of the respondents had completed only their high school diploma, 21.4% finished junior college, 60.7% had a bachelor’s degree, and 13.6% had a doctorate. The normal distribution of the data was verified by examining skewness and kurtosis values. All values fell within the acceptable range of ±2 [74], thereby meeting the assumption of a normal distribution.

### 3.2. Measures

The participants responded to all of the scale items on a seven-point Likert-type scale (1 = not at all, 7 = very much). The survey items for the measured variables are listed in Appendix A. 

*Goal orientation*. The provider’s two different goal orientations were measured using an nine-item scale developed by Brett and VandeWalle [51]. An example of the five-item LGO scale (α = 0.86) is “I often look for opportunities to develop new skills and knowledge”. An example of the four-item PPGO scale (α = 0.83) is “I’m concerned with showing that I can perform better than my coworkers”. 

*Coworker popularity*. Knowledge recipient’s popularity was measured by focal employees who shared knowledge and worked together on a regular basis. An eight-item scale (α = 0.93) developed by Scott and Judge [61] was used to measure coworker popularity. An example is: “My coworker is socially visible”. 

*Knowledge sharing*. The seven-item scale (α = 0.96) developed by Srivastava and his colleagues [75] was utilized to measure knowledge sharing. The coworkers of the focal employees indicated how quantitatively and qualitatively the focal employees perform knowledge-sharing activities. An example is: “Focal employee X shares his/her special knowledge and expertise with me”. 

*Control variables*. Following previous research on knowledge sharing [4,22], we controlled for the following three demographic variables (age, gender, and education level) from the employees. Age was measured in years. Gender (0 = male and 1 = female) and employee education (1 = high school, 2 = junior college, 3 = undergraduate degree, and 4 = graduate degree) were dummy-coded. We also considered tenure with the coworker and task interdependence as crucial factors significantly impacting employee knowledge-sharing behavior. This recognition stems from the heightened interaction among team members when they depend on and collaborate with each other to accomplish their tasks. These two variables were selected as control variables, underscoring their significance in contributing to the accumulation of social capital, acting as an indicator of trust development, and reflecting the developmental stage of the team [7,76]. Tenure with coworker was measured in years, and task interdependence was measured using a five-item scale developed by [77]. An example of the five-item task interdependence scale (α = 0.90) is “I depend on my coworkers’ input to do my job”. While not explicitly posited in the hypothesis formation of this research model, the measurement of PAGO, another dimension of PGO, was undertaken and included as a control variable. An example of the four-item PAGO scale (α = 0.76) is “I prefer to avoid situations at work where I might perform poorly”.

## 4. Results

### 4.1. Confirmatory Factor Analysis

This study conducted a series of confirmatory factor analyses (CFAs) to compare the five-factor baseline model with other alternative models based on chi-square statistics and fit indices of CFI, IFI, TLI, and RMSEA [78]. As reported in Table 1, the hypothesized five-factor model, χ^2^[246] = 762.70, *p* < 0.001; CFI = 0.91; IFI = 0.91; TLI = 0.90, RMSEA = 0.09, fit the data considerably better than any other models. Given the results, there was good discriminant validity of these five measures used in this study.

### 4.2. Descriptive Statistical Analysis

As shown in Table 2, whereas LGO was significantly related to knowledge sharing (*r* = 0.12, *p* < 0.05), PPGO was not significantly related to knowledge sharing (*r* = 0.01, *n.s.*). Coworker popularity (*r* = 0.67, *p* < 0.001) was positively related to knowledge sharing.

### 4.3. Hypothesis Testing

Table 3 shows the series of regression analyses that we used to test the hypothesized effects. While Model 1, which only included control variables, was not statistically significant, we found Model 2, which included goal orientation, to be significant. The explanatory power (R^2^) when two types of goal orientation were added was 5%, and the increment of explanatory power due to the addition of the two variables was also statistically significant (△R^2^ = 0.03, *p* < 0.05). Specifically, the analyses showed a positive association between LGO and knowledge sharing (*β* = 0.22, *p* < 0.01) and a negative association between PPGO and knowledge sharing (*β* = −0.20, *p* < 0.05) (see step 2 in Table 3). In sum, the analyses supported Hypotheses 1 and 2.

In addition, as step 4 in Table 3 shows, our analyses showed that coworker popularity positively moderated the relationship between LGO and knowledge sharing (*β* = 0.13, *p* < 0.05) and negatively moderated the relationship between PPGO and knowledge sharing (*β* = −0.11, *p* < 0.05). According to the procedure proposed by Aiken and West [79], the employee regression lines graphically express the interaction patterns. As shown in Figure 1, the results of our simple slope test revealed that the positive relationship between LGO and knowledge sharing strengthened when coworker popularity was high (*b* = 0.20, *p* < 0.05), but became insignificant when coworker popularity was low (*b* = −0.08, *n.s.*).

Moreover, another simple slope test showed that when coworker popularity was high, the negative relationship between PPGO and knowledge sharing was significant (*b* = −0.18, *p* < 0.05). On the other hand, when coworker popularity was low, the relationship between PPGO and knowledge sharing was not significant (*b* = 0.04, *n.s.*) (see Figure 2). These results support Hypotheses 3 and 4.

## 5. Discussion

### 5.1. Overall Findings

Knowledge sharing can give rise to a social dilemma involving conflict between sharing valuable resources to benefit the whole versus not sharing to protect an individual competitive advantage [6,7]. The findings of this study suggest that the knowledge provider’s tendency to engage in cooperative behaviors and the receiver’s characteristics could play essential roles in attenuating this dilemma. Specifically, we found a positive relationship between LGO and knowledge sharing and a negative relationship between PPGO and knowledge sharing. In addition, our analyses showed that the positive relationship between LGO and knowledge sharing and the negative relationship between PPGO and knowledge sharing intensified when employees perceived coworkers as popular.

### 5.2. Theoretical Implications

Our study has several theoretical implications. First, we explored both sides of knowledge sharing to better understand individual behaviors in the workplace. Since knowledge sharing occurs between individuals in daily interactions, understanding knowledge sharing through the interactions of knowledge providers and recipients is crucial. Nevertheless, the majority of studies on knowledge sharing have predominantly concentrated on knowledge providers, often overlooking the impact of knowledge receivers, with a few exceptions [8,70]. Our results propose that acknowledging the interplay between focal employees and coworkers has the potential to alleviate the knowledge-sharing dilemma. Subsequent research endeavors could further enhance scholarly comprehension of workplace knowledge sharing by building upon our identified dynamics.

Second, this study contributes to goal orientation literature by empirically clarifying and discussing the effects of two dimensions of goal orientation. Efficient knowledge sharing in organizations faces a significant challenge of reconciling the tension between cooperative and competitive forces, as knowledge sharing introduces a social dilemma [6]. By investigating the tension between traditional cooperation and competition (altruism and instrumentalism) along with the explorative and exploitative tendencies exhibited by individuals, this study discerned distinct predictive effects of these two facets of goal orientation [7,80]. Our analysis showed that LGO is closely associated with a preference for cooperation. The benevolent and altruistic tendencies of high-LGO individuals make them perceive their coworkers as allies who enhance their learning by sharing knowledge. In contrast, high-PPGO individuals concentrate on maximizing joint outcomes. Their self-serving negative biases and instrumental motives lead them to regard coworkers as rivals who threaten their superiority. Overall, our results demonstrate that these two goal orientations are critical factors to consider when calculating the costs/benefits of knowledge sharing.

Third, similar to earlier studies [81], our findings strengthen the trait activation theory, emphasizing the moderating influence of coworker characteristics. Crucially, these results are expected to catalyze further theoretical exploration regarding the impact of perceived coworker popularity on goal-achievement processes. This, in turn, influences the degree to which individuals actively pursue and adapt their achievements when confronted with knowledge-sharing dilemmas. We found that LGO individuals tend to cooperate with popular coworkers. These results are consistent with earlier findings that LGO individuals improve task effectiveness in social contexts [31,43]. When faced with popular coworkers, they aim to provide exchange partners with helpful knowledge. Those situated at network cores serve as effective partners, providing prime opportunities for new learning. Engaging with them generates fresh insights and motivates increased resource allocation, even when faced with challenges. Consequently, individuals with these attributes tend to be seen as more desirable collaborators, influencing knowledge-sharing behaviors among employees within LGO. Such benevolent behavior can help coworkers perform at high levels. In the end, based on the principle of reciprocity, LGO benefits the people who initially provide help [6], and this positive reciprocity cycle can significantly benefit entire organizations [2].

On the other hand, people with PPGO have a competitive orientation [13,18,35]. They tend to avoid sharing crucial knowledge with their coworkers because enabling others to benefit from their knowledge might diminish their superiority. Coworkers with higher levels of popularity frequently initiate interpersonal exchanges, drawing attention from others who actively seek their aid upon recognizing their reputations [72]. Consequently, popular coworkers experience heightened concern about preserving their status or reputations. As a result, comparative status assessments among coworkers in the workplace stimulate a competitive mindset among those preferring to protect personal gains over knowledge sharing, causing an overestimation of the associated costs. Thus, interacting with popular coworkers changes this cost/benefit calculation, leading PPGO individuals to overweight the costs of sharing knowledge with popular coworkers. Indeed, previous studies have also shown that PPGO individuals are relatively less constructive than LGO individuals in social situations [57].

Finally, by testing specific coworker characteristics, this study expands the scope of existing research regarding coworkers. As organizations undergo extensive flattening through organizational restructuring, employees’ interactions with their coworkers become more immediate and frequent [26], leading to interdependence and an increased frequency of communication. More than ever, coworkers have become organizations’ most salient resources with whom employees interact [57,58]. Our study provides additional empirical evidence of the importance of coworkers in the workplace.

### 5.3. Practical Implications

This study provides several practical insights into enhancing knowledge-sharing behaviors among employees in organizations. First, considering the differential impact of the two facets of goal orientation on knowledge sharing, the results suggest a rational strategy of considering employees with an altruistic motivation and a perspective oriented towards others, particularly when hiring for roles in important work or team contexts where interpersonal relationships and knowledge sharing are crucial. The study reveals that employees with a learning goal orientation are willing to share knowledge with coworkers despite potential risks, while those with achievement-oriented and self-centered performance prove-goal orientations refrain from sharing knowledge to maintain their competitive edge and superiority. Moreover, in tasks with low interdependence or roles where individual competencies and achievement are critical, individuals with a high achievement prove-goal orientation may be more suitable for staffing, highlighting the applicability of these research findings in workforce allocation. Therefore, it is essential to recognize and consider the differences in individual goal orientations among members when forming teams, giving special attention to teams with a high degree of knowledge utilization.

Second, the study underscores the critical role of specific social contexts in influencing employees’ knowledge sharing and the pivotal function of personal predispositions in such manifestations. Given the increasing interaction among members in the workplace due to the proliferation of team structures and enhanced horizontal communication [26], the findings suggest the need for cultivating favorable relationships with specific colleagues to improve the flow of knowledge among employees. Managers must be aware of the tendency for coworkers to opportunistically exploit social status and relationship-building, taking preventive measures to discourage exploitative behavior and promoting a healthy team culture that encourages positive interactions with preferred coworkers. 

Third, managers should engage in systematic and ongoing monitoring to prevent members from maintaining self-centered behaviors and exclusive relationships with a select few. Continuous encouragement is necessary for members exhibiting altruistic behaviors. These practices, combined with existing organizational recommendations for promoting knowledge sharing, will collectively contribute to the organizational culture. Particularly in the Korean cultural context, where the perception of one’s image through others’ eyes and the importance of collective values are highly valued [82], managers must be mindful of the popularity effect reflecting coworkers’ social status. Efforts should be directed towards the efficient activation of knowledge sharing within this cultural context.

### 5.4. Limitations and Conclusions

This study is not without limitations. First, we cannot arrive at unequivocal conclusions regarding the direction of causality because of the cross-sectional nature of our data. Future longitudinal research could confirm the causal relationships between knowledge sharing and goal orientation/coworker characteristics. Second, as this study exclusively incorporated a restricted set of variables pertaining to individuals and coworkers, future research should explore additional potential factors that could influence the hypothesized relationship. The inclusion of more contextual and relational variables would provide researchers with a more comprehensive understanding of the knowledge-sharing phenomenon within an organizational context. For instance, factors associated with the effective leadership style of supervisors, a cooperative organizational culture, and a climate of learning within teams could serve as significant contextual elements influencing the enhancement of a collaborative team atmosphere. These variables may play a crucial role in altering the relationship between goal orientation and knowledge-sharing behaviors among team members. Third, PAGO remains a valuable approach for global optimization tasks, and its exclusion from this study does not diminish its importance or effectiveness. We acknowledge that future research exploring the effects of PAGO in the context of competition and cooperation optimization problems could provide valuable insights. Fourth, our sample size was smaller than what is typical for studies in the organizational behavior field. Although we found significant interactions that support several of our hypotheses and are confident in our findings, conducting other studies to strengthen the generalizability of our findings could be valuable.

## Figures and Tables

**Figure 1 behavsci-14-00250-f001:**
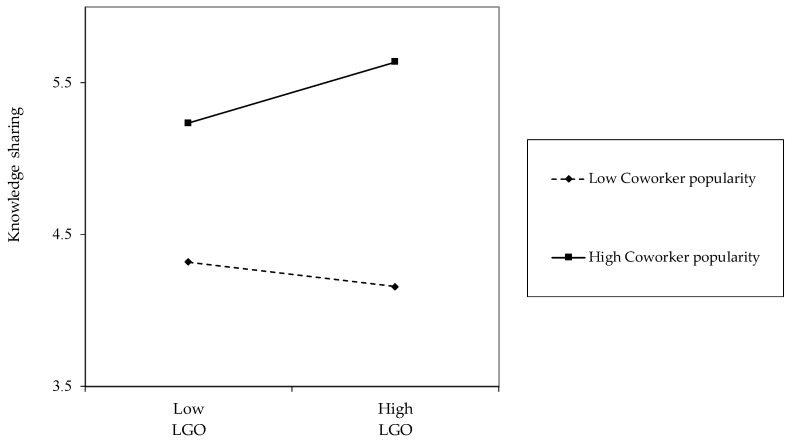
Interaction of learning goal-orientation and coworker popularity based on knowledge sharing.

**Figure 2 behavsci-14-00250-f002:**
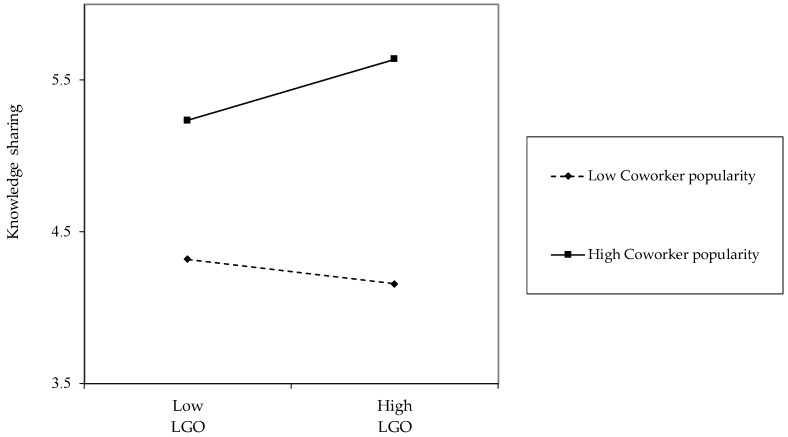
Interaction of performance prove-goal orientation and coworker popularity based on knowledge sharing.

**Table 1 behavsci-14-00250-t001:** Comparison of measurement models.

Model	No. of Factors	χ^2^	df	Δχ^2^	CFI	IFI	TLI	RMSEA
Baseline model	4 factors: LGO, PPGO, CP, KS	762.70	246		0.91	0.91	0.90	0.09
Model 1	3 factors: (LGO + PPGO), CP, KS	997.00	249	234.30 ***	0.87	0.87	0.86	0.11
Model 2	3 factors: LGO, PPGO, (CP + KS)	1766.49	249	1003.79 ***	0.73	0.74	0.71	0.15
Model 3	2 factors: (LGO + PPGO + CP), KS	2033.34	251	1270.64 ***	0.69	0.69	0.66	0.17

*Note*. *** *p* < 0.001. LGO = learning goal-orientation, PPGO = performance prove-goal orientation, CP = coworker popularity, KS = knowledge sharing, CFI = comparative fit index, IFI = incremental fit index, TLI = Turker–Lewis index, RMSEA = root mean square error of approximation.

**Table 2 behavsci-14-00250-t002:** Means, standard deviations, and correlations among study variables.

	Mean	SD	1	2	3	4	5	6	7	8	9	10
1. Age	36.70	8.64										
2. Gender	0.43	0.50	−0.03									
3. Education	2.84	0.71	−0.02	−0.25 ***								
4. Tenure with coworker	3.29	4.14	00.29 ***	−0.06	−0.10							
5. Task interdependence	0.03	1.05	−0.04	−0.01	−0.15 *	−0.10	*(0.90)*					
6. PAGO	3.86	1.02	0.07	0.14 *	0.02	−0.01	0.25 ***	*(0.76)*				
7. LGO	4.75	1.00	0.03	−0.17 **	0.05	0.03	0.04	0.06	*(0.86)*			
8. PPGO	4.32	1.08	0.02	−0.07	0.02	0.01	0.25 ***	0.51 ***	0.55 ***	*(0.83)*		
9. Coworker popularity	5.30	0.93	0.04	−0.02	0.08	0.02	0.06	−0.02	0.14 *	−0.01	*(0.93)*	
10. Knowledge sharing	5.13	0.98	0.06	−0.03	0.09	−0.04	0.07	0.07	0.12 *	0.01	0.67 ***	*(0.96)*

*Note*. *N* = 257, *** *p* < 0.001, ** *p* < 0.01, * *p* < 0.05. PAGO = performance avoid-goal orientation, LGO = learning goal-orientation, PPGO = performance prove-goal orientation. Cronbach’s alpha reliabilities are on the diagonal in parentheses.

**Table 3 behavsci-14-00250-t003:** Hierarchical regression results.

	Knowledge Sharing				
Variables	Model	Model	Model	Model	
	1	2	3	4	VIF
Step 1: Control variables					
Age	0.07	0.07	0.04	0.05	1.12
Gender	−0.01	0.00	−0.02	−0.03	1.15
Education	0.10	0.09	0.03	0.04	1.15
Tenure with coworker	−0.05	−0.05	−0.06	−0.08	1.14
Task interdependence	0.07	0.10	0.02	0.03	1.17
Performance avoid-goal orientation	0.04	0.11	0.11	0.11	1.63
Step 2: Main effects					
Learning goal-orientation (LGO)		0.22 **	0.06	0.06	1.67
Performance prove-goal orientation (PPGO)		−0.20 *	−0.08	−0.07	2.25
Step 3: Moderating variable					
Coworker popularity			0.66 ***	0.67 ***	1.06
Step 4: Moderating effects					
LGO × Coworker popularity				0.13 *	1.18
PPGO × Coworker popularity				−0.11 *	1.23
R^2^	0.02	0.05	0.46	0.48	
R^2^ change		0.03 *	0.41 ***	0.02 *	

*Note*. *N* = 257, *** *p* < 0.001, ** *p* < 0.01, * *p* < 0.05. R^2^ change is the incremental variance explained between each step.

## Data Availability

The data presented in this study are available upon request from the corresponding author.

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
