# Peer review of "Compete or Cooperate? Goal Orientations and Coworker Popularity in the Knowledge-Sharing Dilemma"

_behavsci, 2024, doi:10.3390/bs14030250_

Round 1

Reviewer 1 Report

Comments and Suggestions for Authors

Thanks for the opportunity to review this paper that is very interesting. It is well written in general and presented in a good structure. However, I found some critical components missing in the discussion of hypotheses. While I agree that individual characteristics make a role in the relationship between knowledge sharing and  learning and performing, the organizational culture also has great impact on this relationship, and even more critical. According organizational behavior literature, If the organizational culture motivate knowledge sharing and group goal achievement rather than individual goal achievement, the relationship between knowledge sharing and performance should be positive even with peers popularity. In other words, the relationships proposed in the paper should only represent organizations with aggressive competitive culture. It may not be the case in colleges, research centers, even in project based organizations. The authors totally ignored this group of literature that limited the soundness of the rationale behind hypotheses and therefore failed to provide valid suggestions and implication to organizations. 

Comments on the Quality of English Language

The quality of English is good in my opinion. 

Reviewer 2 Report

Comments and Suggestions for Authors

Title: It reflects the content of the study- okay

Abstract: Encourage the inclusion of specific study findings together with practical and theoretical implications.

Keywords: sufficient

Introduction:  Line 48: some research…, please provide references.

Reviews: Line 231: Why is ‘trait activation theory’ critical for this study? What are other theories referred to before coming to this conclusion? A few lines on this will be interesting. The references are not current, and I cannot see any from 2021 onwards. Please get the latest reviews to ensure the research hypotheses remain relevant and that there is a strong need for the study.

Method:

What is the estimated population size? How the sample size was determined is not clear. Start with research design, population, sample size and distributions, followed by data collection procedures. Was any pre-test and pilot study conducted before the actual data collection? Are all the instruments used free to access? Otherwise, permission to use should be stated – to ensure research ethics.

Findings: The authors use statistics to decide on hypotheses. Good. Encourage to cite some references in the findings.

Implications: please show the gaps in theory and practice and then how the findings of the present study help to fill those gaps. For the practical implication, show how it helps the South Korean companies – be specific to the Korean context. What are the possibilities to generalize the findings?

Conclusion: okay

References: I am not able to see any from 2021 onwards. Appreciate the authors' including the current reviews and update the list.

Comments on the Quality of English Language

Standard English - minor proofreading required 

Reviewer 3 Report

Comments and Suggestions for Authors

Exploring the intricate dynamics of knowledge sharing in the workplace, this article delves into the impact of goal orientations, specifically learning and performance prove-goal orientations, on knowledge-sharing behaviors within employee-coworker dyads, shedding light on the nuanced interplay influenced by coworker popularity as a social cue. There are some suggestions:

1. Please clearly state the practical implications of your findings in the abstract. For example, how can employees and organizations use this information to improve knowledge sharing and work relationships?

2. In the methodology section, please condiser including information on how participant selection was conducted to ensure a representative sample from diverse industries.

3. Please use tables or figures to enhance the readability of your results section.

Reviewer 4 Report

Comments and Suggestions for Authors

The reviewed article raises the current and important issue of the goal orientations and coworker popularity in the knowledge sharing dilemma. Although the issue of knowledge sharing and factors influencing the implementation of this process is not new, on the contrary - it is quite popular and often discussed in the literature on the subject by representatives of various scientific disciplines - the authors have skillfully found a gap in the existing research on this issue.

I perceive the article as scientifically valuable and interesting. In the knowledge-based economy, knowledge sharing is a critical factor determining the success of an organization. Therefore, understanding the essence of this process and exploring the factors influencing its implementation is an important issue both in theoretical and practical aspects.

The hypotheses put forward are well embedded in the analyzed scientific theories.

The strengths of the article include it’s clear and logical structure, an easy-to-read writing style, as well as literature analysis. The selection of the research method and tool as well as the statistical analysis of the obtained data do not raise any doubts. An analysis of the research results was reflected in the discussion and practical conclusions at the end of the article. The authors indicated the limitations of the study and future research directions.

I suggest extending the third part of the article (“Methods”) by providing more information about the study itself. When was the research conducted? Was the selection of companies for the study purposeful?

Next suggestion:

Hypothesis two is: PPGO is negatively related knowledge sharing.

I think it should be: PPGO is negatively related to knowledge sharing.

Summing up, the article is valuable and I congratulate the authors.

Round 2

Reviewer 1 Report

Comments and Suggestions for Authors

I am satisfied with the revision and response to my feedback in the final manuscript. 

Author Response

We are sincerely grateful for the constructive feedback and encouragement that you provided. Your comments are helpful for improving the quality and contribution of this study.